# Efficacy of Unsupervised Self-Collected Mid-Turbinate FLOQSwabs for the Diagnosis of Coronavirus Disease 2019 (COVID-19)

**DOI:** 10.3390/v13081663

**Published:** 2021-08-22

**Authors:** Egildo Luca D’Andrea, Alessia Maria Cossu, Marianna Scrima, Vincenzo Messina, Pasquale Iuliano, Felice Di Perna, Marco Pizza, Fabio Pizza, Nicola Coppola, Luca Rinaldi, Anna Maria Bellizzi, Chiara Pelosi, Carmen Cocca, Angelo Frieri, Fabio Lo Calzo, Giovambattista Capasso, Santina Castriciano, Paolo Maggi, Alessandra Fucci, Michele Caraglia

**Affiliations:** 1COVID Laboratory, Biogem Scarl, Via Camporeale, 83031 Ariano Irpino, Italy; egildoluca.dandrea@biogem.it (E.L.D.); alessiamaria.cossu@biogem.it (A.M.C.); marianna.scrima@biogem.it (M.S.); 2Protein Factory, Biogem Scarl, Via Camporeale, 83031 Ariano Irpino, Italy; 3Molecular Oncology and Precision Medicine Laboratory, Biogem Scarl, Via Camporeale, 83031 Ariano Irpino, Italy; 4Department of Precision Medicine, University of Campania Luigi Vanvitelli, 80138 Naples, Italy; 5Infectious and Tropical Diseases Division, AORN “Sant’Anna and San Sebastiano”, 81100 Caserta, Italy; vincenzo.messina@gmail.com (V.M.); paolo.maggi@unicampania.it (P.M.); 6Respiratory Diseases Division, AORN “Sant’Anna and San Sebastiano”, 81100 Caserta, Italy; pasquale.iuliano@gmail.com (P.I.); fediperna@tin.it (F.D.P.); 7Testami SRL, 80122 Naples, Italy; marco@testami.co (M.P.); fabio.pizza@testami.co (F.P.); 8Department of Mental Health and Public Medicine, University of Campania Luigi Vanvitelli, 80131 Naples, Italy; nicola.coppola@unicampania.it; 9Department of Advanced Medical and Surgical Sciences, University of Campania Luigi Vanvitelli, 80138 Naples, Italy; luca.rinaldi@unicampania.it; 10“S.Ottone Frangipane” Hospital, 83031 Ariano Irpino, Italy; medicinapoariano@aslavellino.it (A.M.B.); docpelosic@gmail.com (C.P.); carmen.cocca@gmail.com (C.C.); info@aslavellino.it (A.F.); fabio.localzo@gmail.com (F.L.C.); 11Translational Nephrology, Biogem Scarl, Via Camporeale, 83031 Ariano Irpino, Italy; gb.capasso@biogem.it; 12Global Scientific Affairs Director at COPAN ITALIA spa Via F. Perotti 10, 25125 Brescia, Italy; santinacm14@gmail.com

**Keywords:** SARS-CoV-2, surveillance, dry swab, unsupervised self-collection

## Abstract

Context: The Global Severe Acute Respiratory Syndrome Coronavirus-2 (SARS-CoV-2) pandemic has resulted in explosive patterns of transmission in most countries. Nasopharyngeal swabs were the specimen’s collection tools recommended for the diagnosis of SARS-CoV-2 infection, and for monitoring infection outbreaks in communities. Our objective was to report the quality and efficacy of unsupervised self-collected mid turbinate “dry FLOQSwabs” (MT FLOQSwabs) (56380CS01, Copan). There were 111 specimens collected for the study: 36 by health care personnel, from themselves, to verify the quality and efficacy of mid-turbinate swabs; 75 to compare and assess the diagnostic performance, among health care personnel, of nasopharyngeal swabs and self-collected mid-turbinate FLOQSwabs. A collection of 51 specimens was enrolled to define the efficacy of the Testami program (validation). Our analyses demonstrate that self-collected mid-turbinate dry swabs ensure an accuracy of 97.3%, as compared to the standard nasopharyngeal swabs collected by health care workers. Furthermore, the mid-turbinate FLOQSwabs can be stored without medium for six days at room temperature without affecting the molecular diagnosis of the SARS-CoV-2 virus infection. Self-collection of diagnostic specimens at home could offer an avenue to increase testing availability for SARS-CoV-2 infection without asking people to travel to a clinic or a laboratory, thus reducing people’s exposure to infection. Our findings demonstrate that unsupervised self-collection swabs, transported dry, are sensitive, practical and easy-to-use tools and should be considered for diagnosis of SARS-COV-2 and coronavirus disease 2019 (COVID-19) surveillance.

## 1. Introduction

In early December 2019, an outbreak of coronavirus disease 2019 (COVID-19), caused by a novel severe acute respiratory syndrome coronavirus 2 (SARS-CoV-2), occurred in Wuhan City, China. Less than one month later, nearly 10 thousand cases were confirmed in China and 82 in 18 other countries, preannouncing the high infectivity of this new coronavirus. Subsequently, several studies confirmed that the average of secondary cases that can be infected by a SARS-CoV-2 positive subject (known as R0 index) is much higher than that of coronaviruses causing SARS and Middle East Respiratory Syndrome (MERS). SARS-CoV-2 has more potential to cause a pandemic, which came true within a short time span [1,2]. Since its discovery, the number of confirmed cases of COVID-19 worldwide exceeds 100 million while the number of global deaths exceeds 2 million. COVID-19 is mainly transmitted by respiratory droplets and aerosol. The incubation period (between contamination and the appearance of the first symptoms) is usually 3–5 days [3]. Early diagnosis of SARS-CoV-2 infection is paramount for disease treatment and infection control. Viral RNA detection by reverse transcription-quantitative real-time polymerase chain reaction (qRT-PCR) on specimens collected by healthcare workers using nasopharyngeal swabs and transported in an appropriate transport medium is the gold standard for diagnosis of SARS-CoV-2 infection [4]. Diagnostic accuracy mainly relies on sample collection appropriateness, specimen stability, and qRT-PCR quality performed on three different viral genes. Copan FLOQSwabs have been already described as capable of preserving RNA of respiratory viruses to be used in molecular analyses without requiring a transport medium [5,6]. Smieja and collaborators demonstrated that nasal mid-turbinated (MT) FLOQSwabs were able to harvest significantly more respiratory epithelial cells than conventional rayon swabs, thus confirming their appropriateness for the diagnosis of infections due to respiratory viruses and suggesting their suitability for self-collection [1,7,8,9]. Moore and colleagues showed that “dry swabs”—not immerged in transport medium for transport and storage—can be used for detecting respiratory viruses with molecular methods, even two weeks after the collection [10]. Finally, a recent publication reported that dry swabs, eluted directly into a simple buffered solution, could support molecular detection of SARS-CoV-2 without compromising assay sensitivity [11]. The main objective of our study was to determine feasibility, acceptability, and diagnostic performances of the developed Testami home self-collection for the SARS-CoV2 RNA testing program, which is based on qRT-PCR detection of SARS-CoV2 RNA in home self-collected specimens by using nasal MT FLOQSwabs transported in a dry tube. To this goal, we firstly evaluated the accuracy of sampling in unsupervised self-collected (SC) and healthcare professional (HCP) collected swabs. Thereafter, the stability of RNA in “dry conditions” was assessed. Finally, diagnostic parameters of the Testami home self-collection program were compared to the standard diagnostic routine for SARS-CoV-2 infection, based on a collection of nasopharyngeal samples by healthcare workers.

## 2. Materials and Methods

### 2.1. Subject and Speciemens

All hospitalized participants with confirmed or suspected COVID-19 infection eligible for study participation were recruited from the infectious diseases department of hospitals: Azienda Ospedaliera “Sant’Anna and San Sebastiano” of Caserta (CE), Universiy of Campania “L. Vanvitelli” and Presidio Ospedaliero “Sant’Ottone Frangipane” of Ariano Irpino (AV) from June through December, 2020. The study was approved by the local ethics committee and conducted according to the Declaration of Helsinki and Good Clinical Practice requirements. Patients presenting for SARS-CoV-2 testing and/or visitors were offered participation in the study. Following the informative notes, the patients gave their written informed consent. A questionnaire assessing the acceptability of self-collection (SC), demographic, and clinical-anamnestic information was provided to patients together with printed instructions on how to collect MT swabs. 

### 2.2. Specimen Collection

In this study, contoured FLOQSwabs w/STOPPER mid-turbinate nasal flocked swabs (MT FLOQSwabs) (56380CS01, Copan) were used (Figure 1).

These swabs have a “stopper”—a collar was positioned 5.5 cm from the tip—as a guide to the maximum insertion depth for adults. The tapered cone-shaped swab allows sampling a large surface area of respiratory mucosa, covering the inferior and middle turbinate bones [1]. Previously, MT swabs were collected at enrollment and six days later, if the patient was still hospitalized by trained bedside nursing staff, placed in eNAT transport media and stored at 2 °C to 8 °C until processed. Subsequently, patients were accompanied to a separate room where they performed self-collection nasal samples with an MT swab as indicated in the instructions and placed the sample in a dry tube without the presence of health personnel in order to simulate home collection. A total of 111 swabs were collected. In order to verify the quality and effectiveness of the swabs of the intermediate turbinates, we analyzed 36 swabs collected by the healthcare staff and by the patients themselves. The remaining 75 swabs were used to compare and evaluate the diagnostic performance between the nasopharyngeal swabs collected by healthcare professionals and the FLOQSwabs of the self-collected middle turbinates. Finally, a new collection of 51 samples was performed to define the effectiveness of the Testami program (Figure 2).

All the samples were double packed and transported to the Biogem laboratory. Upon arrival in the laboratory, they were stored at 2 °C to 8 °C and processed within 24–72 h of sample collection.

### 2.3. RNA Extraction and SARS-CoV-2 Detection

RNA extraction was performed on 200 μL of medium (eNAT) using MagMaxTM viral/pathogen nucleic acid isolation kit (A42352, Thermo Fisher Scientific, Waltham, MA, USA) on KingFisher instrument (Thermo Fisher Scientific, Waltham, MA, USA) according to the manufacturer’s instructions. For dry swabs, sampled cells were resuspended by adding 2 mL of eNAT and vortexing for 5 min. Two hundred microliters of suspension were used for RNA extraction as described before. For the diagnosis of SARS-CoV2 infection, the qRT-PCR was performed on 5 μL of eluted RNA using TaqPath™ COVID-19 PCR kit multiplex qRT-PCR test (A48102, Thermo Fisher Scientific, Waltham, MA, USA) according to the manufacturer’s instructions. The viral targets analyzed were ‘S’ gene, ‘N’ gene, and ‘ORF1ab’ gene. Negative and positive controls were run simultaneously with samples. For each analysis, MS2 phage RNA was added as control of extraction and amplification procedures. The result was considered positive when cycle threshold (Ct) values were <37 for at least two out of three SARS-CoV-2 target genes. As additional control on sample adequacy, the housekeeping human Beta globin gene was analyzed on the extracted RNA, using a different multiplex assay kit SARS-CoV-2, Novel Coronavirus 2019-nCoV Nucleic Acid Detection Kit CE-IVD (Bioer Technology Co., Ltd., Hangzhou, China) according to manufacturers’ instructions. The beta-globin cut-off of cycle threshold (Ct) values were 35. While this kit detects two viral genes (ORF1ab and N), we used TaqPath™ COVID-19 PCR kit for diagnosis of SARS-CoV-2 because of internal validated standard operating procedures. 

### 2.4. SARS-CoV-2 Virus Stability In Vitro and in Self-Collected MT Dry Swab

In order to have preliminary data on SARS-CoV-2 virus RNA stability in SC swabs stored and transported without transport medium, a trial stability test was performed. To this goal, residual transport medium (Universal Transport Medium, UTM) from SARS-CoV-2 negative (*N* = 2) and SARS-CoV2 positive (*N* = 2) swabs were used. Negative and positive samples were separately pooled, centrifuged to enrich cellular material, and resuspended in 400 µL of UTM. The obtained cell suspensions were used to soak 16 MT swabs. The swabs were dried for a few minutes under the laminar hood and then stored for 2, 4, and 6 days, at 4 °C or room temperature (22–28 °C). At the completion of each incubation, the swabs were placed in 1mL of eNAT medium, vortexed for 30 s and stored at −20 °C until the analyses. RNA extraction was performed on 200 μL of eNAT medium. qRT-PCR analyses were performed using both TaqPath™ COVID-19 PCR kit, and SARS-CoV-2 Novel Coronavirus 2019-nCoV Nucleic Acid Detection Kit according to the manufacturer’s instructions. The stability of RNA in SC samples was also evaluated in a small cohort of volunteers (*N* = 5). A reference nasopharyngeal swab was collected to assess SARS-CoV-2 infection in the enrolled subjects: Two of them were identified as COVID-19 positive, and three as COVID-19 negative. Each subject self-collected two samples with MT swab. One SC sample was immediately placed in an eNAT medium (t0). The second SC sample was kept at room temperature (18–25 °C) for six days before proceeding with RNA extraction and analysis, performed as described above.

### 2.5. Evaluation of Diagnostic Performances of the “Testami Home Self-Collection Program”

In order to evaluate the diagnostic performances of the proposed project, subjects with a current SARS-CoV2 infection (*N* = 40) and healthy subjects (*N* = 11) were enrolled in the study. The assembled self-collection kit (Testami kit), together with the instructions for use, was provided to each subject. The kit included a swab package containing a single MT swab and a screw-top container without medium, a zip lock sample bag, and a tertiary container. The kit was picked up from the collection site and brought to the Biogem laboratory within 24–48 h from the collection. Viral genes and beta-globin levels were assessed as described before to identify samples with no human tissue presence. Notably, none of the tested samples failed the beta-globin positive control.

### 2.6. Statistical Analysis

All analyses were performed using PRISM GraphPad v 7.0. Concordance between SC and HCP swabs was calculated using Cohen’s Kappa (κ), which measures agreement between the categorical assignments given by two methods. The statistic takes values typically between zero and one. A κ > 0.80 indicates very good agreement, while κ = 1 indicates perfect concordance. Cycle threshold (Ct) values were recorded for all positive test results and mean Ct was compared between SC and HCP collected samples. Sensitivity and specificity of the self-collection method were compared with HCP-obtained swabs and CI was calculated using the Wilson–Brown method. *p* values were computed by Fisher’s Exact test. Spearman’s and Pearson’s tests were used to assess the correlation of the PCR Ct values across the tested groups.

## 3. Results

All the participants in the study found the unsupervised self-collection procedure easy and did not experience adverse events. Furthermore, unsupervised self-collection was preferred by most of the participants, likely because with SC the subjects can control the comfort level of nasal collection better than a trained healthcare professional. The capability of home MT nasal swabs to ensure an adequate self-collection of the biological specimen was evaluated by assessing beta-globin RNA, used as human cellular housekeeping control for evaluating samples’ quality. Preliminary studies were performed using only nasal MT FLOQSwabs for specimen collection, performed either by volunteers (self-collected-SC) or healthcare professionals (HCP); quantitative real-time PCR analyses of beta-globin confirmed that MT FLOQSwabs are suitable for self-collection (data not shown), as reported in the literature [1]. To compare sampling efficacy of self-collected MT FLOQSwabs (SC) to that of healthcare worker (HCP)-collected standard nasopharyngeal swabs, qRT-PCR analysis of beta-globin was performed. Beta-globin was detected (Ct ≤ 35) in 104 SC samples and 100 HCP samples, with a Ct mean of 28.93 (CI95%: 28.26–29.61) in SC and 29.15 (28.37–29.94) in HCP standard swabs (Figure 3A). 

The agreement between the two collection methods, in terms of sampling efficacy, was evaluated using Cohen’s Kappa (κ); the percentage of agreement was 97.3%, with κ = 0.9459. According to these results, home self-collected MT swabs were considered adequate for the collection of biological samples. The statistical correlation in detecting SARS-CoV2 infection between SC and HCP swabs was assessed in *N* = 52 COVID-19 positive patients and *N* = 23 healthy subjects (*p* value = 0.0075) (Table 1).

As shown in (Appendix A), the analysis revealed that sensitivity and specificity of self-collected MT FLOQSwabs (SC) vs. standard nasopharyngeal swabs collected by trained personnel (HCP) were 94.23% (CI 95%: 84.36–98.43) and 95.65% (CI95%: 79.01 to 99.78), respectively. The Positive Predictive Values (PPVs) and Negative Predictive Values (NPVs) were 98.00% (CI95%87.80–99.70) and 88.00% (CI95% 70.0–95.67), respectively. Considering the Ct values for the three target genes (average), there was a good correlation between SC and HCP swab (Spearman correlation test, r = 0.8431 (CI95% 0.7591 0.8995), *p* < 0.0001) (Figure 3B), confirming that SC swabs are valuable tools for diagnosis. Preliminary in vitro results indicated that SARS-CoV2 RNA collected with the MT swab and stored without medium is stable up to six days at room temperature without affecting the viral genes’ amplification quality. These results were also confirmed by analyzing home self-collected dry swabs stored for six days at RT: Beta-globin was detected in all the tested samples, and two of them were correctly identified as SARS-CoV2 positive (Appendix A).

This result is in agreement with those obtained from the standard diagnostic procedures performed on the same samples (data not shown). The analyses were then performed on the swabs that underwent the whole Testami program from the delivery of the home self-collection kit to the SARS-CoV2 RNA analyses in Biogem laboratories (Figure 4A).

The high correlation of the PCR Ct values across the two groups was demonstrated by Spearman’s test: r =0.8608 (CI95%: 0.7685 to 0.9180), *p* value < 0.0001 (Figure 4B). The results obtained confirmed a significant correlation between the standard diagnostic program (eNAT swab) and the Testami program as shown in Table 2 (*p* value = 0.0002). The developed Testami program ensures high sensitivity (95%, CI95% 83.5–99.11) and specificity 90.91 (CI95% 62.26–99.53). The Positive Predictive Values (PPVs) and Negative Predictive Values (NPVs) were 97.44% (CI95% 86.82–99.87) and 83.33% (CI95% 55.2–97.04), respectively (Appendix A). Overall, with this study, we demonstrated that specimens for SARS-CoV2 infection diagnosis can be efficiently home self-collected by MT swabs and can be safely stored without medium for at least six days without reducing the reliability of molecular diagnosis (Appendix A). The proposed program, based on home self-collection and molecular analysis in an authorized laboratory, will increase the accessibility to a correct diagnosis and will represent an easy and robust alternative to traditional swab collection, with an important impact on the global health crisis.

## 4. Discussion

Accurate diagnosis of SARS-COV-2 viral infection certainly depends on effective collection of samples before proceeding with the diagnostic method. The gold standard diagnostic test for SARS-CoV-2 infection is the detection of viral RNA by quantitative real-time reverse transcription polymerase chain reaction (RT-qPCR), given its sensitivity, specificity, and speed. To date, the best alternative to sampling collection is a storage of the flocked swab in a specific means of transport. This study aimed to demonstrate that the measurement of the cellularity of the “dry” unsupervised self-collected middle turbinate flocked nasal swab was effective in assessing the validity of the viral load. We performed molecular cell quantification of homemade MT nasal swabs by evaluating beta-globin RNA by qRT-PCR. Beta-globin was used as a human cell control to assess the quality of the samples. The results indicated that SARS-CoV-2 RNA collected with MT buffer and stored without a medium was stable for up to six days at room temperature without affecting the amplification quality of viral genes. The power of our study is limited in terms of the number of individuals enrolled, but the reported results certainly offer new useful information to ensure easy access to the correct diagnosis. However, while further studies on larger populations are needed to reinforce our conclusions, we suggest that an unsupervised self-collection swab would be useful for the diagnosis and surveillance of COVID-19.

## 5. Conclusions

Our findings demonstrate that unsupervised self-collection swabs, transported dry, are sensitive, practical and easy-to-use tools and should be considered for diagnosis of SARS-COV-2 and coronavirus disease 2019 (COVID-19) surveillance.

## Figures and Tables

**Figure 1 viruses-13-01663-f001:**
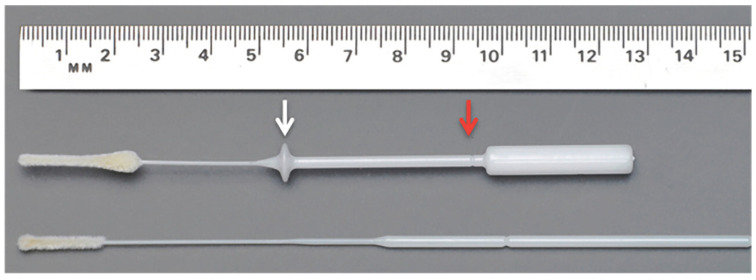
Contoured FLOQSwabs w/STOPPER mid-turbinate nasal flocked swabs (MT FLOQSwabs) (56380CS01, Copan) (**top**) compared to flocked nasopharyngeal FLOQSwabs. Adapted from Smieja et al. (**bottom**). Blank arrow indicates the “stopper”; red arrow indicates the break point.

**Figure 2 viruses-13-01663-f002:**
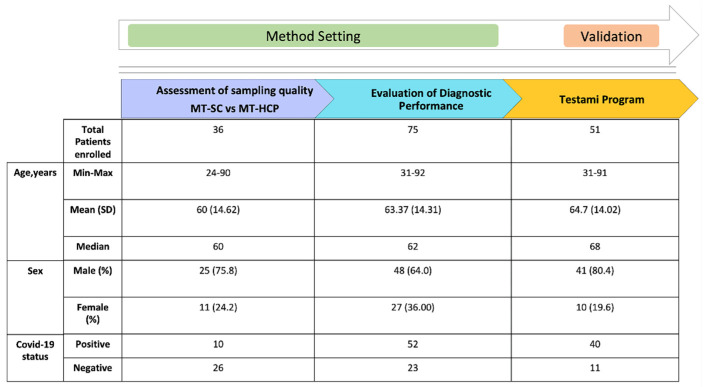
Profile of subjects enrolled for the different phases of the study.

**Figure 3 viruses-13-01663-f003:**
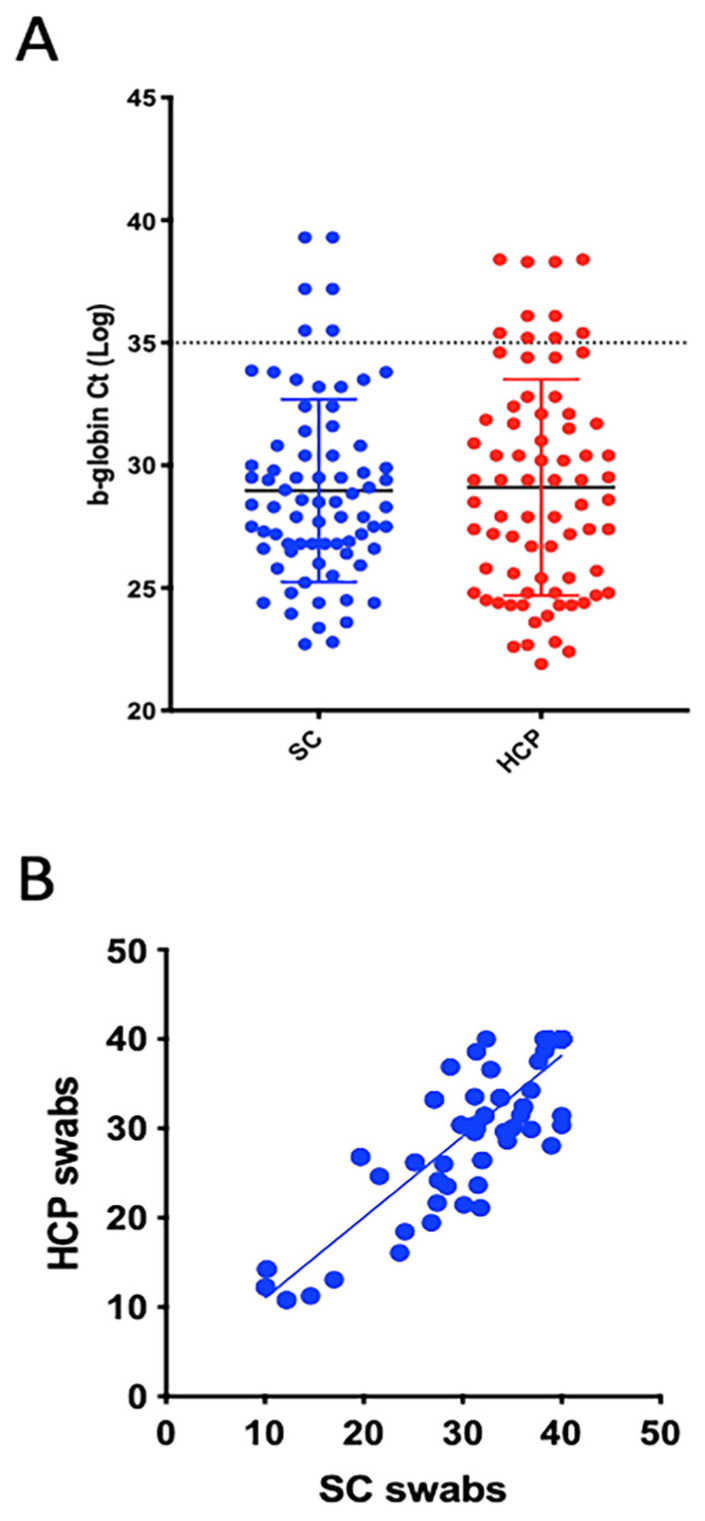
Graphical representation of SC and HCP swabs correlation. (**A**) Scatter plot showing the b-globin levels detected in self-collected (SC) and healthcare worker-collected (HCP) samples. Note that Ct values are above the cut-off value (35 Ct) in seven SC samples and 11 HCP samples. (**B**) Correlation of Ct values of viral genes (average) detected in SC and HCP swabs (*N* = 75). Pearson’s correlation test, r = 0.8784 (CI 95% 0.8136–0.9216), *p* < 0.0001.

**Figure 4 viruses-13-01663-f004:**
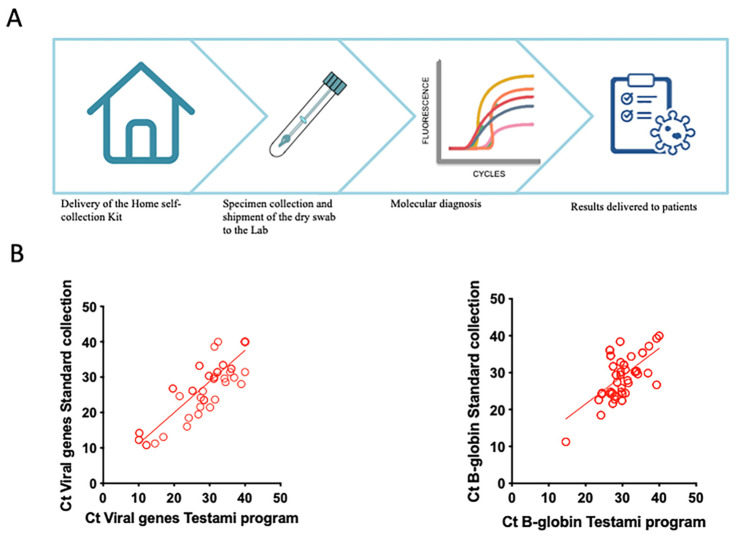
Representation and validation of Testami program. (**A**) Schematic representation of Testami program. (**B**) Correlation of Ct values of viral genes in sample collected using the Testami program and by standard procedures, Spearman’s test r = 0.8608 (CI95%: 0.7685 to 0.9180), *p* value < 0.0001 (**left** panel); correlation of Ct values of B-globin gene in samples collected using the Testami program and by standard procedures (Spearman’s test r = 0.4318 (CI95%: 0.2889 to 0.5559), *p* value < 0.0001 (**right** panel).

**Table 1 viruses-13-01663-t001:** Statistical correlation between the SC Swab and the standard diagnostic HCP Swab.

		HCP Swabs			
		**Total**	**COVID-19 Positive**	**COVID-19 Negative**	***p*-Value**
**SC Swabs**	**COVID-19 Positive**	*50*	*49*	*1*	<0.05
	**COVID-19 Negative**	*25*	*3*	*22*	
	**Total**	*75*	*52*	*23*	

SC Swab: Self-collected MT FLOQSwabs. HCP Swab: Nasopharyngeal swabs collected by trained personnel. *p* value < 0.05. The statistical correlation was determined using Chi square test.

**Table 2 viruses-13-01663-t002:** Statistical correlation between the standard diagnostic program (eNat swab) and the Testami program.

		Standard Diagnostic Program (eNat Swab)			
		**Total**	**COVID-19 Positive**	**COVID-19 Negative**	***p*-Value**
**Testami program**	**COVID-19 Positive**	*39*	*38*	*1*	<0.05
	**COVID-19 Negative**	*12*	*2*	*10*	
	**Total**	*51*	*40*	*11*	

*p* value < 0.05. The statistical correlation was determined using Chi square test.

## Data Availability

All the data related to the present study are stored at Biogem and are accessible if required.

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
