# Peer review of "Efficacy of Unsupervised Self-Collected Mid-Turbinate FLOQSwabs for the Diagnosis of Coronavirus Disease 2019 (COVID-19)"

_viruses, 2021, doi:10.3390/v13081663_

Round 1
Reviewer 1 Report
There are many English spelling and grammatical errors. Please change the following:
Line 58: "to cause a pandemic"
Line 61: "transmitted by respiratory droplets and aerosols"
Line 66: "gold standard"
Line 90: "Specimens"
Line 116: "performed"
Line 156: not sure what the authors are trying to saying with "immerging 16 MT swabs". Emerging is the correct spelling, but this word still doesn't make sense to me.
Line 157: "dried for a few minutes"
Line 179: "none of the tested samples failed the beta-globin positive control"
Line 192: "all the participants"
Line 193 "did not experience"
Line 208: "HCP" is shown as "HPC" in y-axis lable
Line 229: non-English style use of commas and periods "98.00". Same problem occurs in legend for Figure 4, Scheme 1, Suppl Table 2, Suppl Table 3, Suppl Table 4.
Line 232: "tools"
Line 279: what is a "human cell cleaning control"?
Line 280: "In particular." Incomplete sentence needs to be fixed.
Line 285: sentence needs to be fixed.
Other concerns that need to be addressed:
Scheme 1 is referred to as Supplemental Table 1 on line 225.
There is no reference to Suppl Table 2 anywhere in the Results or Discussion.
The number of numerical digits included in Suppl Table 4 should be reduced to match the rest of the manuscript.
No data are shown or explained to support the statement made on lines 192-4.
The text in Figure 2 is too small to see. Shouldn't this be called a Table instead of a Figure?
On Line 143, it is stated that the Ct cutoff value used was "<37". In Figure 3A, it is shown to be 35. Which is it?
There are no units shown on the y-axis for Figure 3A.
Why is the spacing different for the x-axis and the y-axis on Figure 3B/4B? For example, the space from 0 to 10 on the x-axis is larger than the same space from 0 to 10 on the y-axis. The reader is expecting to see a 45 degree angle on the trendline if there is a strong correlation, but that can't work when the spacing is off.
It's not clear to me why the authors included both Fig 3B and Fig 4B. When I read Fig 3B, I was hoping to see a trendline shown with the slope indicated. Then I found the data re-shown in Fig 4B with a trendline.
Were data points above 35 Ct (or is it 37?) included in the Spearman correlation test analysis for Fig 3B/4B? They are shown in the graph, but in the text it says that such results were not considered to be positive, so it would make the most sense to exclude them I would think.
I cannot figure out what Tables 1 and 2 are trying to show. Above the column headings it says "HCP swabs" and alongside the row headings it says "SC swabs", and there is only one number shown where the column and the row intersect. Were the values always exactly the same for each type of sample (other areas of the paper indicate that they were not the same)? If not, shouldn't there be a value for HCP and another one for SC?
Author Response
We express our gratitude to the Referees for their comments that will be useful in order to improve our manuscript. We have considered all the concerns appropriately raised and we have tried to address all of them as specified below.
In the revised version of our manuscript, all changes are marked using the “Word's Track Changes” tool.
Query 1: There are many English spelling and grammatical errors. Please change the following:
Response 1: We thank the Referee for putting our attention to the need to change English spelling and grammatical errors. Wemodified following the order indicated for each line.
- On line 156 we change the word “ immerging “ with “sooking” to improve the sense.
- On line 229, in legend of Figure 4, Suppl Table2, Suppl Table 3 and Suppl Table 4 we replaced the comma with the point.
Query 2: Other concerns that need to be addressed:
Response 2:
- In the supplementary tables, we modified “Scheme 1 “ with “ Supplemental Table 1”
- The reference to Suppl Table 2 is mentioned on line 237.
- We reduced the number of numerical digits included in Suppl Table 4.
- We remove in the text “ According to the answers to the provided questionnaire” not essential for testing purposes.
- We increased the test in the figure 2. We decided to call Figure 2 and not Table 2, because it is not a classic table, but graphic elements have been added to make it clearer.
- On line 143 the Ct cutoff value for viral genes analysis was “<37” to analyzed according to the TaqPath™ COVID-19 PCR kit multiplex qRT-PCR test manufacturer’s instructions. In Figure 3A, we showed 35 as cut off to analyzed Beta-globin gene according to SARS-CoV-2, Novel Coronavirus 2019-nCoV Nucleic Acid Detection Kit CE-IVD (Bioer Technology Co., Ldt., Hangzhou, China) assay kit instructions.
- On Figure 3A we added the units on the y-axis.
Query 3: Why is the spacing different for the x-axis and the y-axis on Figure 3B/4B? For example, the space from 0 to 10 on the x-axis is larger than the same space from 0 to 10 on the y-axis. The reader is expecting to see a 45 degree angle on the trendline if there is a strong correlation, but that can't work when the spacing is off.
Response 3 :
In the figure 3A and figure 4B we made the space equal on the x-axis and y-axis.
Query 4: It's not clear to me why the authors included both Fig 3B and Fig 4B. When I read Fig 3B, I was hoping to see a trendline shown with the slope indicated. Then I found the data re-shown in Fig 4B with a trendline.
Response 4:
We thank the Referee for putting your attention on this mistake.
As introduced in the manuscript we used 75 specimens to compare and assess diagnostic performance among health care personnel nasopharyngeal swabs and self-collected mid-turbinate FLOQSwabs and a collection of 51 specimens to define the efficacy of Testami program.
In Figure 3B we analyzed the correlation of Ct values of viral genes detected in SC and HCP swabs, while in Figure 4B (uploaded in the manuscript the exact 4B figure) we analyzed the correlation of Ct values of viral genes in 51 sample collected to validate Testami program. Moreover, we modified Fig 3B with a trendline as shown in Figure 4B.
Query 5: Were data points above 35 Ct (or is it 37?) included in the Spearman correlation test analysis for Fig 3B/4B? They are shown in the graph, but in the text it says that such results were not considered to be positive, so it would make the most sense to exclude them I would think.
Response 5:
In Figure 3B and 4B we showed a data below the cut off of Ct 37, because we used TaqPath™ COVID-19 PCR kit multiplex qRT-PCR test to analyzed Ct viral genes.
Query 6: I cannot figure out what Tables 1 and 2 are trying to show. Above the column headings it says "HCP swabs" and alongside the row headings it says "SC swabs", and there is only one number shown where the column and the row intersect. Were the values always exactly the same for each type of sample (other areas of the paper indicate that they were not the same)? If not, shouldn't there be a value for HCP and another one for SC?
Response 6:
Table 1 and 2 are 2X2 table showed statistical correlation. To make more understandable we changed the statistical analyses using the McNemar test.
We thank again the referee for the useful suggestions, and we hope that now the revised version of our manuscript can be considered for publication.
Best Regards,
Alessandra Fucci, PhD
and
Michele Caraglia, MD, PhD
Full Professor of Biochemistry

Reviewer 2 Report
This is a comparison of self-collected nasal mid-turbinate swab with professionally-collected nasopharyngeal, indicating similar high sensitivity. There is also an analysis of the stability of medium-free, dry swabs, potentially enabling home self-collection and sending to a laboratory facility.
- The high sensitivity of nasal mid-turbinate swabs confirms previous papers, but adds corroborating data.
- The stability of dry swabs is novel and interesting for simplifying home collection and sending to a laboratory.
- The three described populations, and how many people/swabs were done, is extremely confusing. On reading the paper twice, I was confused as to how many swabs were done and on whom. The paper states 111 swabs, yet there were 162 swabs/people described (36 health care plus 75 plus 51 in Testami) This data (people and number of swabs) should be added to the table and shoudl be described more clearly.
- Figure 1 and 2 have used inappropriate statistical analyses. A 2X2 table should use the McNemar test for paired data, with a P-value of >0.05 indicating no difference.
Author Response
We express our gratitude to the Referees for their comments that will be useful in order to improve our manuscript. We have considered all the concerns appropriately raised and we have tried to address all of them as specified below.
In the revised version of our manuscript, all changes are marked using the “Word's Track Changes” tool.
Query 1: The three described populations, and how many people/swabs were done, is extremely confusing. On reading the paper twice, I was confused as to how many swabs were done and on whom. The paper states 111 swabs, yet there were 162 swabs/people described (36 health care plus 75 plus 51 in Testami) This data (people and number of swabs) should be added to the table and shoudl be described more clearly.
Response 1:
In total we performed 162 swabs; in particular 111 swabs are collected for method setting: 36 by health care personnel and from themselves to verify quality and efficacy of mid-turbinate swabs and 75 to compare and assess diagnostic performance among health care personnel nasopharyngeal swabs and self-collected mid-turbinate FLOQSwabs. A collection of 51 specimens is enrolled to validate to efficacy of Testami program.
Query 2: Figure 1 and 2 have used inappropriate statistical analyses. A 2X2 table should use the McNemar test for paired data, with a P-value of >0.05 indicating no difference.
Response 2:
We modified the statistical analyses in 2X2 table using the McNemar test with a P-value of >0.05.
We thank again the referee for the useful suggestions, and we hope that now the revised version of our manuscript can be considered for publication.
Best Regards,
Alessandra Fucci, PhD
and
Michele Caraglia, MD, PhD
Full Professor of Biochemistry

Round 2
Reviewer 1 Report
I recommend for acceptance in the present form.
Author Response
Dear Reviewer,
thank you for your recommendation. We revised English language and style.
Reviewer 2 Report
A little bit clearer than originally.
Line 160, "soaking" is misspelled.
Table 1. This is not a McNemar test result. I calculate P=0.617 (not significant), which indicates the two are comparable.
Table 2. This is not a McNemar test result. I calculate P=1.000 (not significant). Again, this indicates the two tests are comparable.
Line 262--3 significant digits after the decimal would be enough.
Author Response
Dear Reviewer,
thanks for your precious insight on our manuscript.
Query 1:
Line 160, "soaking" is misspelled.
Response 1:
We modified "soaking" with "to soak"
Query 2:
Table 1. This is not a McNemar test result. I calculate P=0.617 (not significant), which indicates the two are comparable.
Table 2. This is not a McNemar test result. I calculate P=1.000 (not significant). Again, this indicates the two tests are comparable.
Response 2:
Thank you for pointing out the mistake, we performed a statistic chi square test, significant with pvalue lower than 0.05.
Query 3:
Line 262--3 significant digits after the decimal would be enough.
Response 3:
